# Anxiety and associated factors among Ethiopian health professionals at early stage of COVID-19 pandemic in Ethiopia

Henok Dagne[1]*, Asmamaw Atnafu[2], Kassahun Alemu[3], Telake Azale[4], Sewbesew Yitayih[5], Baye Dagnew[5], Abiy Maru Alemayehu[6], Zewudu Andualem[1], Malede Mequanent Sisay[3], Demewoz Tadesse[7], Soliyana Hailu Chekol[8], Eyerusalem Mengistu Mamo[8], Wudneh Simegn[9]

1 Department of Environmental and Occupational Health and Safety, Institute of Public Health, College of Medicine and Health Sciences, University of Gondar, Gondar, Ethiopia, 2 Department of Health System and Policy, Institute of Public Health, College of Medicine and Health Sciences, University of Gondar, Gondar, Ethiopia, 3 Department of Epidemiology and Biostatistics Institute of Public Health, College of Medicine and Health Sciences, University of Gondar, Gondar, Ethiopia, 4 Department of Health Education and Behavioral Sciences, College of Medicine and Health Sciences, University of Gondar, Gondar, Ethiopia, 5 Department of Psychiatry, School of Medicine, College of Medicine and Health Sciences, University of Gondar, Gondar, Ethiopia, 6 Department of Optometry, School of Medicine, College of Medicine and Health Sciences, University of Gondar, Gondar, Ethiopia, 7 Ethiopian National Blood Bank Service, Addis Ababa, Ethiopia, 8 Ghandi Memorial Hospital, Addis Ababa, Ethiopia, 9 Department of Pharmaceutics, School of Pharmacy, University of Gondar, Gondar, Ethiopia

* enoch2313@gmail.com

## Abstract

### Introduction

In late 2019, a new coronavirus disease known as COVID-19 (novel coronavirus disease 2019) was identified. As there is no any drug to treat this pandemic, the healthcare professionals are disproportionately at higher risk. The mental health outcome is expected to be high. Anxiety is expected to have a significant impact on health professionals, especially among those who work without adequate resources for self-protection.

### Objectives

The objectives of this research was to assess self-reported anxiety symptoms and associated factors among Ethiopian healthcare professionals in the early stages of the pandemic.

### Methods

We have conducted an online cross-sectional study to collect information from healthcare professionals in Ethiopia during the early stage of the outbreak from April 7, 2020 to May 19, 2020. GAD-7 was used for measurement of anxiety. We have used a cut of point of 10 and above to report anxiety symptoms. We have used Google Forms for online data collection and SPSS-22 for analysis. To determine associated factors for anxiety, a binary logistic regression model was used. Variables with p-value < 0.2 during the bivariable binary logistic regression were exported for further analysis in the multivariable binary logistic regression.

**Data Availability Statement:** All relevant data are within the manuscript and its Supporting information files.

**Funding:** No funding agent was involved in this study.

**Competing interests:** The authors have declared that no competing interests exist.

**Abbreviations: AOR**, Adjusted Odds Ratio; **CI**, Confidence Interval; **COR**, Crude Odds Ratio; **EPI Info**, Epidemiological Information; **SPSS**, Statistical Package for Social Sciences.

Finally, variables with p-value <0.05 were considered as significantly associated with the outcomes.

## Results

Three hundred and eighty-eight healthcare professionals filled the online questionnaire; Majority (71.1%) were males. Significant number of respondents (78.9%) reported lack of adequate personal protective equipment (PPE) at the work place. The prevalence of anxiety was 26.8%.

Being female (AOR: 1.88; 95% C.I:1.11, 3.19), visiting/treating 30–150 patients per day (AOR: 3.44; 95% C.I:1.51, 7.84), those employed at private healthcare institutions (AOR: 2.40; 95% C.I:1.17, 4.90), who do not believe that COVID-19 is preventable (AOR: 2.04; 95% C.I:1.04, 4.03) and those who reported lack of PPE (AOR: 1.98; 95% C.I:1.04, 3.79) were more likely to be anxious.

## Conclusions

The anxiety prevalence among healthcare professionals in Ethiopia during early stage of COVID-19 pandemic was high. This study shows that lack of preventive equipment, being female, contact with many patients, low self-efficacy and working in private health facilities were risk factors for anxiety. Anxiety prevention among health professionals during COVID-19 pandemic requires a holistic approach including provision of sufficient PPE, improving self-efficacy and addressing problems both at public and private institutions and focusing more on female health professionals.

## Introduction

The Coronavirus Disease 2019 (COVID-19) epidemic emerged in Wuhan, China, and spread to other countries [1]. COVID-19 is a cluster of acute respiratory illness characterized by fever, cough, myalgia or fatigue, pneumonia, dyspnea, headache, diarrhea, hemoptysis, runny nose, and phlegm- producing cough [2, 3]. Because of the sudden nature of the outbreak and the infectious power of the virus, it will inevitably cause serious threats to people's physical health and lives. It has also triggered a wide variety of psychological problems, such as anxiety [1, 4, 5]. The COVID-19 pandemic has been attributed to a range of anxiety and mental illness exacerbation [6, 7]. A recent umbrella review showed that the prevalence of anxiety among healthcare workers during the COVID-19 pandemic was 24.94% [8]. Another systematic review also showed that the pooled prevalence of anxiety was 30% [9]. An anxiety prevalence of 19.8% was reported among frontline and second line healthcare workers in Italy [10]. The overall prevalence of anxiety disorder among the healthcare workers in Nepal during the COVID-19 pandemic was 37.3% [11]. The prevalence of anxiety among China's healthcare workers before the peak time of COVID-19 was 40% [12]. A prevalence of 46.7% anxiety was reported among health workers in Libya during the civil war and COVID-19 pandemic [13]. About 54.2% of healthcare workers in China had symptoms of anxiety [14]. Another study conducted among healthcare workers on the frontlines in Egypt and Saudi Arabia revealed that 58.9% of study participants had symptoms of anxiety [15]. Several predisposing factors are identified for anxiety during COVID-19. These factors include excessive working hours [16], lack of/insufficient personal protective equipment [17, 18], contact with patients with

suspected COVID-19 [19], age, department, years of experience, working hours per week, internal displacement, stigmatization [13], being married, not living alone [20], and female sex [21–25].

Anxiety affects the outcome of chronic diseases such as diabetes, cardiovascular diseases, cancer, and obesity [26]. It also affect performance of job, quality of sleep, routine activities, and productivity of the affected individual [27].

Although pandemics including COVID-19 can trigger a significant human toll as well as public anxiety, economic loss, and other adverse consequences, it is common for health practitioners and administrators to fully concentrate on disease prevention and care, abandoning the psychological and mental consequences secondary to the case. As a result, there is a deficit of coping mechanisms, which increases the burden of related diseases [28].

Understanding and researching the psychological conditions of health workers during this turbulent period is therefore crucial. As a result, the objective of this study is to assess prevalence of anxiety and its contributing factors among healthcare professionals during the early stages of the COVID-19 outbreak. This research will provide evidence for tailoring and implementing appropriate mental health care policies to effectively deal with the outbreak's challenges.

## Methods and materials

### Study design, area and period

We adopted cross-sectional study design through online survey, to assess symptoms of anxiety and associated factors among health care professionals in Ethiopia. The survey was conducted from April 7, 2020 to May 19, 2020.

### Population and inclusion criteria

All health care professionals living in Ethiopia and who were Ethiopian nationals were used as study population. We included all health care professionals who were social media (Facebook, Twitter, Email and Instagram) users, and who were volunteer to fill the survey. We have excluded health professionals who had no access to internet during the study period due to different reasons. We preferred to use social media users because it enables us to collect the data without direct contact with the study participants, which is crucial to reduce the rate of spread of the COVID-19 pandemic.

### Sample size determination and sampling technique

Due to the lack of published literature investigating the anxiety during the pandemic COVID -19 in Ethiopia as well as in the study area, in the present study, we calculated the maximum possible sample size. To achieve this, 50% proportion, 5% margin of error, a 95% confidence level and 5% for the non-response rate was considered during the sample size calculation. The final sample size was 404. The snow ball sampling technique was used to access health care professionals.

### Data collection instrument and measurement

We used generalized anxiety disorder 7 items scale to assess level of anxiety [29, 30]. The GAD-7 scale was reliable with Cronbach's alpha of 0.92 and test re-test reliability (intra class correlation = 0.83) with good validity [31]. Questions related to socio-demographic information were incorporated. Participants were asked how often they were bothered by each symptom during the last 2 weeks. Response options were "not at all," "several days," "more than half

the days," and "nearly every day," scored as 0, 1, 2, and 3, respectively. A score of 10 or greater represents a reasonable cut point for identifying cases of anxiety as explained elsewhere [31]. Similar cut of point was also used previously [17]. Even though there are different tools to assess anxiety, we have used GAD-7 as it has been found to have great psychometric properties and is short and easy to administer [32].

## Data processing and analysis

As this was an online data collection in the form of CSV (excel file), there was no need of data entry. The excel form data were imported into SPSS version 22 for analysis. All assumptions for binary logistic regression were checked. Bi-variable and multivariable logistic regressions were computed to determine predictor variables for symptoms of anxiety. Variables with a p-value <0.2 during the bivariable binary logistic regression analysis were included in the multivariable logistic regression analysis. In multivariable binary logistic regression, variables were considered as significant at a $p$-value of < 0.05. Hosmer and Lemeshow goodness- of -fit test (p>0.05) was used to check model fitness. Descriptive and inferential statistics were performed.

## Ethical approval and consent to participate

Ethical approval was obtained from the Ethical Review Committee of Environmental and Occupational Health and Safety department, the University of Gondar. Respondents were communicated via social media. After explaining the purpose of the study, respondents were asked to fill and submit their responses. Any potential identifiers were eliminated to ascertain confidentiality.

## Results

A total of 388 health professionals with a response rate of 96% participated in the study. Majority (71.1%) were males. Nine out of ten of the participants were public (government employees). Above half (53.1%) of the participants were medical doctors. Majority (88.1%) believe that COVID-19 is preventable. Most of the health professionals (69.1%) live at least with one family member. Significant number of respondents (78.9%) reported lack of sufficient personal protective equipment (PPE) at the work place (Table 1).

## Prevalence of anxiety among health professionals in Ethiopia during an early stage of COVID-19 pandemic

The prevalence with 95% confidence intervals of anxiety among health professionals in Ethiopia during an early stage of COVID-19 pandemic was 26.8% (22.4%, 30.9%).

## Factors associated with anxiety among health professionals in Ethiopia during an early stage of COVID-19 pandemic

Sex, living with at least one family member, average number of patients visited per day, organizational affiliation, whether professionals think that COVID-19 is preventable and sufficient availability of PPE were candidate variables (with p-value<0.2) during the bivariable binary logistic regression. Except living with at least one family member, all of these variables were significantly associated with anxiety among healthcare professionals in the final multivariable binary logistic regression model. Female study participants were 1.88-folds (AOR: 1.88; 95% C.I:1.11, 3.19) at higher adjusted odds of developing anxiety as compared to males. Healthcare professionals who visited above 30 patients per day were 3.44-times (AOR: 3.44; 95% C.I:1.51,

**Table 1. Socio-demographic characteristics of health professionals screened for anxiety symptoms in Ethiopia during an early stage of COVID-19 pandemic (n = 388).**

| Variables | Category | Frequency (n) | Percent (%) |
|---|---|---|---|
| Sex | Male | 276 | 71.1 |
| | Female | 112 | 28.9 |
| Age in years | 23–26 | 92 | 23.7 |
| | 27–28 | 92 | 23.7 |
| | 29–31 | 103 | 26.5 |
| | 32–55 | 101 | 26.0 |
| Educational status | Diploma/degree | 233 | 60.1 |
| | MSc and above | 155 | 39.9 |
| Current marital status | Married | 159 | 41.0 |
| | Unmarried | 229 | 59.0 |
| Work experience | Junior (0–2 years) | 127 | 32.7 |
| | Mid-level (3–5 years) | 117 | 30.2 |
| | Senior (>5 years) | 144 | 37.1 |
| Organizational affiliation | Governmental | 347 | 89.4 |
| | Private | 41 | 10.6 |
| Average patients per day (n = 387) | ≤9 | 71 | 18.3 |
| | 10–19 | 122 | 31.5 |
| | 20–29 | 92 | 23.8 |
| | ≥30 | 102 | 26.4 |
| Profession | Human medicine | 206 | 53.1 |
| | Other health science | 182 | 46.9 |
| Living with at least one family member | No | 120 | 30.9 |
| | Yes | 268 | 69.1 |
| Sufficient PPE availability | No | 306 | 78.9 |
| | Yes | 82 | 21.1 |
| Do you think that COVID-19 is preventable? | No | 46 | 11.9 |
| | Yes | 342 | 88.1 |

7.84) at higher odds of developing anxiety than those who visited less than an average of nine patients per day. Health professionals employed at private healthcare institutions were 2.4 times (AOR: 2.40; 95% C.I:1.17, 4.90) more likely to be anxious compared to those working at public healthcare institutions. Study subjects who do not believe that COVID-19 is preventable were 2.04 times (AOR: 2.04; 95% C.I:1.04, 4.03) and those who reported lack of PPE were 1.98 times (AOR: 1.98; 95% C.I:1.04, 3.79) more likely to develop anxiety (Table 2).

## Discussion

Anxiety was found to be present in 26.8% of the population with 95% confidence intervals (22.4%, 30.9%). Female healthcare staff, those who saw a higher number of patients per day, those who worked in private healthcare facilities, health professionals who believe COVID-19 is not preventable, and those who indicated a lack of personal protective equipment (PPE) at work were more likely to experience anxiety symptoms.

The prevalence of anxiety in the current study is higher than several earlier study reports from China [33–36] and Italy [10] and lower than other study reports from the China [12, 14, 37–39], Nepal [11], Libya [13], and Egypt and Saudi Arabia [15]. The disparity may be attributed to differences in the anxiety assessment instrument used, the prevalence of COVID-19

**Table 2. Factors associated with anxiety among health professionals in Ethiopia during an early stage of COVID-19 pandemic (n = 388).**

| Variables | | Anxiety | | COR 95% CI | AOR 95% CI |
|---|---|---|---|---|---|
| | | No (%) | Yes (%) | | |
| Sex | Female | 72(25.4) | 40(38.5) | 1.84(1.14,2.96) | 1.88(1.11,3.19)* |
| | Male | 212(74.6) | 64(61.5) | 1 | 1 |
| Presence of family member living with | No | 82(28.9) | 38(36.5) | 1.42(0.88,2.28) | 1.42(0.86,2.36) |
| | Yes | 202(71.1) | 66(63.5) | 1 | 1 |
| Average number of patients visited per day | ≤9 patients | 62(21.9) | 9(8.7) | 1 | 1 |
| | 10–19 patients | 84(29.7) | 38(36.5) | 3.12(1.40,6.92) | 2.22(0.97,5.11) |
| | 20–29 patients | 71(25.1) | 21(20.2) | 2.04(0.87,4.78) | 1.89(0.80,4.50) |
| | 30–150 patients | 66(23.3) | 36(34.6) | 3.76(1.67,8.43) | 3.44(1.51,7.84)* |
| Organization | Public | 260(91.5) | 87(83.7) | 1 | 1 |
| | Private | 24(8.5) | 17(16.3) | 2.12(1.09, 4.12) | 2.40(1.17,4.90)* |
| Do you think that COVID-19 is preventable? | No | 26(9.2) | 20(19.2) | 2.36(1.26,4.45) | 2.04(1.04,4.03)* |
| | Yes | 258(90.8) | 84(80.8) | 1 | 1 |
| PPE availability | No | 218(76.8) | 88(84.6) | 1.66(0.91,3.03) | 1.98(1.04,3.79)* |
| | Yes | 66(23.2) | 16(15.4) | 1 | 1 |

1 = Reference group,

* Significant at p < 0.05,

** Significant at p < 0.001, Hosmer and Lemeshow goodness of fit test (p = 0.338).

and the cut-off values used to dichotomize the outcome. However, the current prevalence of anxiety was similar with a pooled prevalence from a systematic review reported by Sofia et al. [40], a study among medical staff in a tertiary infectious disease hospital for COVID-19 [41], a report from a recent umbrella review [8] and another systematic review [9].

Healthcare professionals with higher patient load were more likely to be anxious. This is not surprising as the number of patients visited increases, healthcare professionals' risk to COVID-19 becomes high. Healthcare practitioners will encounter shortage of time to exercise COVID-19 preventive practice as the number of patients they have to see per day is higher than the maximum standard.

Changing gloves and washing hands after each patient visit would be extremely difficult for them, especially if the resources available to them at work are limited.

Health professionals working at private settings were more likely to become anxious. No earlier study has reported the disparity in anxiety among public and private healthcare institutions so far as to our extensive literature search. However, we believe that the relative freedom to stay at home whenever possible from public healthcare setup and the strict attendance from private healthcare institutions may explain the higher anxiety level among healthcare professionals working at private settings. Further qualitative study may be needed to explore the real reasons of this discrepancy.

Study subjects who reported lack of PPE were more likely to be anxious in the current study. Similar to this finding a study in Hong Kong revealed that respondents who were more bothered by not having enough surgical masks were more likely to have poor mental health [42]. Depletion of PPE is known to contribute to psychological distress [38]. During the Ebola outbreak, many health workers worked extra-hours and settings without personal protective equipment and driven mainly by compassion resulted in mental health problems disproportionately higher than the general public [43]. A study on psychological impact and coping strategies of frontline Medical staff in Hunan, China revealed that the availability of personal protective equipment provided psychological benefits [44].

In the current study female health professionals were more likely to report anxiety as compared to males. This is in line with several previous studies conducted to see the gender difference of anxiety prevalence [1, 21–24, 40, 44–50]. Several possible explanations have been given for higher level of anxiety disorder among females as compared to their counter parts. Studies [51, 52] have reported that the female reproductive cycle may contribute to the significantly higher prevalence of anxiety in women. The intensive fluctuations in oestrogen and progesterone during the menstrual cycle, pregnancy, or postpartum periods were related to changes in the hormone's neuroprotective effects, which might increase the chronicity correlated with anxiety occurrence [51]. A study also related the lower risk of developing anxiety in males to differential access to appropriate health services [53]. A metacognitive beliefs in uncontrollability, advantages and avoidance of worry may also contribute to the higher prevalence of anxiety in females than males [54]. So far, several environmental, genetic and physiological factors were suggested that may play a significant role in the differences between females and males in anxiety development [55–57]. A study also showed that women were more reactive than men in neural networks associated with fear and arousal responses [58]. The high prevalence of anxiety among female health professionals in Ethiopia may also be due to higher family responsibilities culturally bestowed on women. Women are usually involved in highly strenuous home activities in addition to their job at work place. A study conducted among 23 countries showed that females reported higher levels of anxiety compared to men at the aggregate data [50]. The study [50] also revealed that in some countries, there was no sex difference in the anxiety level reported. Similarly, there was no significant difference based on gender in a study conducted in China [59]. This might be due to the fact that GAD-7 rates vary by ethnic/cultural group [60].

Finally, health professionals who believe that COVID-19 is not preventable were more likely to be anxious as compared to those who believed that it can be prevented. The lack of hope in prevention of the disease may be indicative of the level of anxiety the healthcare workers are facing.

## Limitation of the study

This cross-sectional study is based on self-reported data. Its sampling design is susceptible to bias as it is from internet-based surveys, and the sample does not really represent the general population. The social desirability bias is also another limitation of this study. However, the study is useful to the country for intervention as early as possible to halt the mental health impact of COVID-19 among healthcare professionals.

## Conclusions

Anxiety was prevalent among healthcare professionals in Ethiopia during the early stage of COVID-19 pandemic. Patient load, lack of PPE and working in private institutions were factors for anxiety. Females and those who believed that COVID-19 prevention is impossible were more likely to be anxious. The healthcare institutions should fulfill necessary supplies of PPE and establish mental health units to deal with the prevalent cases of anxiety.

## Supporting information

**S1 File.**
(SAV)

## Acknowledgments

The authors are grateful for study participants, University of Gondar, individuals and associations who helped in dissemination of the data collection tool.

## Author Contributions

**Conceptualization:** Henok Dagne, Asmamaw Atnafu, Kassahun Alemu, Telake Azale, Sewbesew Yitayih, Baye Dagnew, Abiy Maru Alemayehu, Zewudu Andualem, Malede Mequanent Sisay, Demewoz Tadesse, Eyerusalem Mengistu Mamo, Wudneh Simegn.

**Data curation:** Henok Dagne.

**Formal analysis:** Henok Dagne, Wudneh Simegn.

**Investigation:** Henok Dagne, Asmamaw Atnafu, Kassahun Alemu, Telake Azale, Baye Dagnew, Abiy Maru Alemayehu, Zewudu Andualem, Malede Mequanent Sisay, Demewoz Tadesse, Eyerusalem Mengistu Mamo, Wudneh Simegn.

**Methodology:** Henok Dagne, Asmamaw Atnafu, Kassahun Alemu, Telake Azale, Sewbesew Yitayih, Baye Dagnew, Abiy Maru Alemayehu, Zewudu Andualem, Malede Mequanent Sisay, Soliyana Hailu Chekol, Wudneh Simegn.

**Project administration:** Henok Dagne, Baye Dagnew, Abiy Maru Alemayehu, Zewudu Andualem, Soliyana Hailu Chekol, Eyerusalem Mengistu Mamo, Wudneh Simegn.

**Resources:** Henok Dagne, Baye Dagnew, Zewudu Andualem, Demewoz Tadesse, Soliyana Hailu Chekol, Eyerusalem Mengistu Mamo, Wudneh Simegn.

**Software:** Henok Dagne.

**Supervision:** Henok Dagne.

**Visualization:** Henok Dagne.

**Writing – original draft:** Henok Dagne.

**Writing – review & editing:** Henok Dagne, Asmamaw Atnafu, Kassahun Alemu, Telake Azale, Sewbesew Yitayih, Baye Dagnew, Abiy Maru Alemayehu, Zewudu Andualem, Malede Mequanent Sisay, Demewoz Tadesse, Soliyana Hailu Chekol, Eyerusalem Mengistu Mamo, Wudneh Simegn.

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
