## [Decision Letter · Decision Letter 0]

11 Dec 2020

PONE-D-20-32653

Anxiety and Associated Factors among Ethiopian Health Professionals at Early Stage of COVID-19 Pandemic in Ethiopia

PLOS ONE

Dear Dr. Dagne Derso,

Thank you for submitting your manuscript to PLOS ONE. After careful consideration, we feel that it has merit but does not fully meet PLOS ONE’s publication criteria as it currently stands. Therefore, we invite you to submit a revised version of the manuscript that addresses the points raised during the review process.

An expert in the field handled your manuscript, and we are grateful for their time and contributions. Although some interest was found in your study, numerous major concerns arose that must be addressed in your revised manuscript. Please respond to ALL of the reviewer's comments.

We look forward to receiving your revised manuscript.

Kind regards,

Frank T. Spradley

Academic Editor

PLOS ONE

"The funders had no role in study design, data collection and analysis, decision to publish, or preparation of the manuscript"

6. We noticed you have some minor occurrence of overlapping text with the following previous publication(s), which needs to be addressed:

- https://www.nature.com/articles/srep28033

- https://www.researchsquare.com/article/rs-34504/v1

The text that needs to be addressed involves the Abstract, Background, and Discussion specifically. In your revision ensure you cite all your sources (including your own works), and quote or rephrase any duplicated text outside the methods section. Further consideration is dependent on these concerns being addressed.

Reviewers' comments:

Reviewer's Responses to Questions

**Comments to the Author**

1. Is the manuscript technically sound, and do the data support the conclusions?

Reviewer #1: Partly

2. Has the statistical analysis been performed appropriately and rigorously? 

Reviewer #1: Yes

3. Have the authors made all data underlying the findings in their manuscript fully available?

Reviewer #1: Yes

4. Is the manuscript presented in an intelligible fashion and written in standard English?

Reviewer #1: No

5. Review Comments to the Author

Reviewer #1: Anxiety and Associated Factors among Ethiopian Health Professionals at Early Stage of COVID-19 Pandemic in Ethiopia

This manuscript is important as it assesses anxiety and associated factors among health professionals during COVID-19 which is crucial for developing an intervention plan that will address the problem. However, the introduction, methods, result, and discussion parts of the manuscript need more improvement. Also, the focuses of this study is not clear because the authors sometimes talk about depression the other time about anxiety in the introduction, method, and discussion part. So, this manuscript confuses its reader in its current status. Moreover, there are many grammatical errors that need to be edited by a language expert or a native language speaker.

Abstract

Line 44: “Developing the outcome” is not clear for readers. Try to make the term you use as simple as possible to reach all audiences.

Line 53: It would be better if you leave reporting 95% CI.

Line 54: What do you mean when you say “a large number of patients”? Do you mean treating many patients? It looks confusing.

Line 58-68: what are the implications of your findings?

Main document

Introduction

Line 67: Either delete this sentence or make it clear for readers.

Line 72 and 73: It would be better to delete depression and focus on the topic of your study.

Line 65-86: Less focus was given to anxiety. I was expecting detailed information regarding COVID-19 related anxiety but found very few sentences related to anxiety among health professionals. My recommendation is to make more literature reviews and come up with a strong introduction.

Material and Methods

Line 88-92: Information about study area is missed.

Line 90: Delete the sampling technique and report it under the topic of “sample size determination and sampling technique”.

Line 95: It would be better if you report the specific social media you have used to collect data.

Line 93-96: The inclusion and exclusion criteria are not reported well.

Line 98-102: The sampling technique is not clearly mentioned, and the reason why the authors used the snowball technique is not reported. If the authors have used social media, I do not think the snowball sampling technique is the appropriate method, but your justification is important.

Line 104-111. There are varieties of questionnaires to assess anxiety but there is no clear information regarding why you selected GAD-7. Also, the specificity, sensitivity, and validity of this questionnaire in the study area and other places are not mentioned.

Line 110-111: Use reference regarding the cut-off.

Line 116: If you have analyzed the data for anxiety and depression why you did not change the topic of your study?

Result

Line 131-132, table 1: What do you mean “other health professionals”?

Discussion

Line 160-161: it is a repetition of what you wrote in the introduction. So, try to rewrite this part.

Line 166 -1167: It is better if you mention the place or countries where the previous studies were done including whether it was conducted among a similar population or not.

Line 167-168: What do you mean about the difference in the study population? Also, you were mentioning depression several times including here. So, what you have assessed in this study is not clear, is that depression or anxiety?

Line 171: it is better to leave comparing your study against population-based studies.

Line 197-201: It would be better to find other more convincing explanations regarding why females have higher anxiety during this pandemic.

Line 207-208: Why your finding is different from the study conducted in China. It is better if you discuss the differences.

6. PLOS authors have the option to publish the peer review history of their article (what does this mean?). If published, this will include your full peer review and any attached files.

Reviewer #1: No

---

## [Author Response · Author response to Decision Letter 0]

11 May 2021

Authors’ responses 

Title: Anxiety and Associated Factors among Ethiopian Health Professionals at Early Stage of COVID-19 Pandemic in Ethiopia

Manuscript Number: PONE-D-20-32653

Dear Editor/reviewers, we are very grateful for your valid comment given regarding our revised manuscript that helped us to improve the write up. We have considered your eminent comments and suggestions and modified the manuscript accordingly. The response is indicated as authors’ response next to each issues raised. 

Thank you for submitting your manuscript to PLOS ONE. After careful consideration, we feel that it has merit but does not fully meet PLOS ONE’s publication criteria as it currently stands. Therefore, we invite you to submit a revised version of the manuscript that addresses the points raised during the review process.

An expert in the field handled your manuscript, and we are grateful for their time and contributions. Although some interest was found in your study, numerous major concerns arose that must be addressed in your revised manuscript. Please respond to ALL of the reviewer's comments.

Authors’ response: Thank you, we have gone through all the issues raised and addressed all of them accordingly.

1. Please ensure that your manuscript meets PLOS ONE's style requirements, including those for file naming. The PLOS ONE style templates can be found https://journals.plos.org/plosone/s/file?id=wjVg/PLOSOne_formatting_sample_main_body.pdf and https://journals.plos.org/plosone/s/file?id=ba62/PLOSOne_formatting_sample_title_authors_affiliations.pdf

Authors’ response: Thank you, we have followed the formatting requirements of plos one as indicated

2. Please provide additional details regarding participant consent. In the ethics statement in the Methods and online submission information, please ensure that you have specified (1) whether consent was informed and (2) what type you obtained (for instance, written or verbal, and if verbal, how it was documented and witnessed). If your study included minors, state whether you obtained consent from parents or guardians. If the need for consent was waived by the ethics committee, please include this information. If you are reporting a retrospective study of medical records or archived samples, please ensure that you have discussed whether all data were fully anonymized before you accessed them and/or whether the IRB or ethics committee waived the requirement for informed consent. If patients provided informed written consent to have data from their medical records used in research, please include this information.

Authors’ response: Thank you, our study did not include minors as the legal age for employment is 18 years in the country. We have mentioned that the consent obtained is written informed consent in the revised manuscript. Participants were requested to sign electronically if they wish to participate in the survey. 

"The funders had no role in study design, data collection and analysis, decision to publish, or preparation of the manuscript"

a. Please clarify the sources of funding (financial or material support) for your study. List the grants or organizations that supported your study, including funding received from your institution.

d. If you did not receive any funding for this study, please state: “The authors received no specific funding for this work.”

Authors’ responses: No funding agent was involved in this study and we have included this statement in the revised cover letter. 

Authors’ response: Thank you, we have included the data as a supplementary file in the revised submission. 

Authors’ response: No legal restriction or sensitive patient information is included in the current study.

Authors’ response: We have uploaded the data set. 

Authors’ response: Thank you, we have moved it to the method section. 

6. We noticed you have some minor occurrence of overlapping text with the following previous publication(s), which needs to be addressed:

- https://www.nature.com/articles/srep28033

- https://www.researchsquare.com/article/rs-34504/v1

The text that needs to be addressed involves the Abstract, Background, and Discussion specifically. In your revision ensure you cite all your sources (including your own works), and quote or rephrase any duplicated text outside the methods section. Further consideration is dependent on these concerns being addressed.

Authors’ response: we have gone through the manuscript word for word and we have rephrased, cited and improved the write-up as suggested.

Reviewers' comments:

Reviewer #1: Anxiety and Associated Factors among Ethiopian Health Professionals at Early Stage of COVID-19 Pandemic in Ethiopia

This manuscript is important as it assesses anxiety and associated factors among health professionals during COVID-19 which is crucial for developing an intervention plan that will address the problem. However, the introduction, methods, result, and discussion parts of the manuscript need more improvement. Also, the focuses of this study is not clear because the authors sometimes talk about depression the other time about anxiety in the introduction, method, and discussion part. So, this manuscript confuses its reader in its current status. Moreover, there are many grammatical errors that need to be edited by a language expert or a native language speaker.

Authors’ response: Thank you for the issues raised, we have corrected the write-up and edited for editorial problems. We have meticulously corrected problems at introduction, methods, result, and discussion parts as suggested by the reviewer. 

Abstract

Line 44: “Developing the outcome” is not clear for readers. Try to make the term you use as simple as possible to reach all audiences.

Authors’ response: Thank you we changed it as “being anxious” 

Line 53: It would be better if you leave reporting 95% CI.

Authors’ response: Thank you, we changed it as suggested.

Line 54: What do you mean when you say “a large number of patients”? Do you mean treating many patients? It looks confusing.

Authors’ response: Thank you we meant treating many patients and we have specifically mentioned it as treating/visiting 30-150 patients per day. 

Line 58-68: what are the implications of your findings?

Authors’ response: we have included the following statement as an implication and revised the conclusion section “This study shows that lack of preventive equipment, being female, contact with many patients, low self-efficacy and working in private health facilities were risk factors for anxiety. Anxiety prevention among health professionals during COVID-19 pandemic requires a holistic approach including provision of sufficient PPE, improving self-efficacy and addressing problems both at public and private institutions and focusing more on female health professionals.” 

Main document

Introduction

Line 67: Either delete this sentence or make it clear for readers.

Authors’ response: we have revised it. 

Line 72 and 73: It would be better to delete depression and focus on the topic of your study.

Authors’ response: Thank you, we have changed it. 

Line 65-86: Less focus was given to anxiety. I was expecting detailed information regarding COVID-19 related anxiety but found very few sentences related to anxiety among health professionals. My recommendation is to make more literature reviews and come up with a strong introduction.

Authors’ response: Thank you, we have added additional literature about anxiety among health professionals during COVID-19 pandemic. 

Material and Methods

Line 88-92: Information about study area is missed.

Authors’ response: we have added additional information about the study area.

Line 90: Delete the sampling technique and report it under the topic of “sample size determination and sampling technique”.

Authors’ response: Thank you, we have deleted the sampling technique mentioned in the study design, area and period section and mentioned it in the sample size determination section. 

Line 95: It would be better if you report the specific social media you have used to collect data.

Authors’ response: Thank you very much we have used social media such as Facebook, Twitter, Email and Instagram

Line 93-96: The inclusion and exclusion criteria are not reported well. 

Authors’ responses: Thank you, we have mentioned the following in the revised manuscript 

“We have excluded health professionals who had no access to internet during the study period due to different reasons. We have used. We preferred to use social media users because it enables us to collect the data without direct contact with the study participants, which is crucial to reduce the rate of spread of the COVID 19 pandemic.”

Line 98-102: The sampling technique is not clearly mentioned, and the reason why the authors used the snowball technique is not reported. If the authors have used social media, I do not think the snowball sampling technique is the appropriate method, but your justification is important.

Authors’ responses: Thank you we have requested our study participants to send the questionnaire link to their closest friend so that he/ she will participate in the study. This is due to two basic reasons. The first is due to the busy hours they have health professionals do not usually participate in online surveys easily which may result in low response rate, the second reason is that they are more likely respond to questions when they are approached by their closest friends. Because of these reasons, we have used a targeted social networking even though we have shared the questionnaire link over social media. 

Line 104-111. There are varieties of questionnaires to assess anxiety but there is no clear information regarding why you selected GAD-7. Also, the specificity, sensitivity, and validity of this questionnaire in the study area and other places are not mentioned.

Authors’ response: Thank you, we have mentioned these informations in the revised manuscript in the data collection and measurement section. 

Line 110-111: Use reference regarding the cut-off.

Authors’ response: Thank you we have included reference for the cut of points. 

Line 116: If you have analyzed the data for anxiety and depression why you did not change the topic of your study? 

Authors’ response: Thank you for this comment, we have corrected the typos error throughout entire write-up and focused only on anxiety. 

Result

Line 131-132, table 1: What do you mean “other health professionals”? 

Authors’ response: we meant that health professionals other than medicine which includes Nurses, midwives, optometry, environmental health etc. 

Discussion

Line 160-161: it is a repetition of what you wrote in the introduction. So, try to rewrite this part.

Authors’ response: thank you, we have removed the mentioned statement. 

Line 166 -1167: It is better if you mention the place or countries where the previous studies were done including whether it was conducted among a similar population or not.

Authors’ response: Thank you, the countries and the population can be clearly observed from the references cited. However, we have mentioned the place and population as requested by the reviewer. 

Line 167-168: What do you mean about the difference in the study population? Also, you were mentioning depression several times including here. So, what you have assessed in this study is not clear, is that depression or anxiety? 

Authors’ response: Thank you, we have removed the term depression and we meant by difference in the study population as the population are different based on the facilities and the countries they are working in, the level of support they receive and the living standard they have as well as other socioeconomic factors even though they are both health professionals. We have rewritten this section to remove ambiguities as we have removed the population based studies as suggested by the reviewer in the forth-coming comment. 

Line 171: it is better to leave comparing your study against population-based studies.

Authors’ response: Thank you, we have removed the population based study as suggested. 

Line 197-201: It would be better to find other more convincing explanations regarding why females have higher anxiety during this pandemic.

Authors’ responses: we have searched additional evidences for the difference in the levels of anxiety among female health professionals and we have added the additional possible explanations. However, we believe that this study can never give a lasting solution to the different views regarding the difference as there are contradicting evidences. This requires a meta-analysis and additional holistic primary evidences. 

Line 207-208: Why your finding is different from the study conducted in China. It is better if you discuss the differences.

Authors’ responses: we have mentioned the possible difference in the revised manuscript.

---

## [Editor Report · Decision Letter 1]

20 May 2021

Anxiety and Associated Factors among Ethiopian Health Professionals at Early Stage of COVID-19 Pandemic in Ethiopia

PONE-D-20-32653R1

Dear Dr. Dagne,

We’re pleased to inform you that your manuscript has been judged scientifically suitable for publication and will be formally accepted for publication once it meets all outstanding technical requirements.

Kind regards,

Frank T. Spradley

Academic Editor

PLOS ONE

---

## [Editor Report · Acceptance letter]

31 May 2021

PONE-D-20-32653R1 

Anxiety and Associated Factors among Ethiopian Health Professionals at Early Stage of COVID-19 Pandemic in Ethiopia 

Dear Dr. Dagne:

I'm pleased to inform you that your manuscript has been deemed suitable for publication in PLOS ONE. Congratulations! Your manuscript is now with our production department. 

Kind regards, 

on behalf of

Dr. Frank T. Spradley 

Academic Editor

PLOS ONE